# A Hybrid Recommender System Based on Autoencoder and Latent Feature Analysis

**DOI:** 10.3390/e25071062

**Published:** 2023-07-14

**Authors:** Shangzhi Guo, Xiaofeng Liao, Gang Li, Kaiyi Xian, Yuhang Li, Cheng Liang

**Affiliations:** 1College of Computer Science, Chongqing University, Chongqing 400044, China; 20211401018g@cqu.edu.cn (S.G.); xfliao@cqu.edu.cn (X.L.); 20211401019g@cqu.edu.cn (G.L.); 20211401021g@cqu.edu.cn (K.X.); 2College of Computer and Information Science, Southwest University, Chongqing 400715, China; limouhang@email.swu.edu.cn; 3Institute of Artificial Intelligence and Blockchain, Guangzhou University, Guangzhou 510006, China

**Keywords:** data science, deep neural network, Latent Feature Analysis, multi-metric recommender system, matrix representation

## Abstract

A recommender system (RS) is highly efficient in extracting valuable information from a deluge of big data. The key issue of implementing an RS lies in uncovering users’ latent preferences on different items. Latent Feature Analysis (LFA) and deep neural networks (DNNs) are two of the most popular and successful approaches to addressing this issue. However, both the LFA-based and the DNNs-based models have their own distinct advantages and disadvantages. Consequently, relying solely on either the LFA or DNN-based models cannot ensure optimal recommendation performance across diverse real-world application scenarios. To address this issue, this paper proposes a novel hybrid recommendation model that combines Autoencoder and LFA techniques, termed AutoLFA. The main idea of AutoLFA is two-fold: (1) It leverages an Autoencoder and an LFA model separately to construct two distinct recommendation models, each residing in a unique metric representation space with its own set of strengths; and (2) it integrates the Autoencoder and LFA model using a customized self-adaptive weighting strategy, thereby capitalizing on the merits of both approaches. To evaluate the proposed AutoLFA model, extensive experiments on five real recommendation datasets are conducted. The results demonstrate that AutoLFA achieves significantly better recommendation performance than the seven related state-of-the-art models.

## 1. Introduction

In the current era characterized by abundant information, individuals are confronted with a deluge of extensive data [1,2,3,4]. Notable examples include the colossal amount of data generated by Google, reaching the scale of petabytes, and Flickr, which produces terabytes of data on a daily basis [5,6]. The challenge at hand is to devise an intelligent system capable of extracting relevant information from these vast datasets [7,8,9]. One practical approach to tackle this challenge is the utilization of a recommender system (RS). RSs play crucial roles in enhancing online services, contributing to both business growth and improved user experiences [10]. Typically, a user-item rating matrix is employed to capture user preferences across various items such as news, short videos, music, movies, and commodities [11]. In this matrix, each row represents a specific user, each column corresponds to a specific item, and each entry signifies a user’s preference for a particular item [3]. The key to implementing an RS lies in uncovering users’ latent preferences for different items based on this user-item rating matrix [12,13].

Numerous approaches have been proposed for implementing an RS. Among them, the Latent Feature Analysis (LFA) model has gained significant popularity in industrial applications due to its efficiency and scalability [14]. When applied to a user-item rating matrix, the LFA model projects users and items onto a shared low-dimensional Latent Feature space [15]. By training two low-dimensional matrices using the observed entries only [16], the LFA model can estimate the missing entries by leveraging these trained matrices [17,18,19,20]. As a result, the LFA model offers advantages in terms of efficiency and scalability, particularly in industrial contexts. However, it should be noted that the LFA model is a linear model and may not effectively address complex non-linear relationships between users and items [21].

In recent times, the rapid advancement of deep learning has led to the widespread adoption of deep neural networks (DNNs) [22,23,24] in RSs [25,26]. DNNs have emerged as a promising approach for capturing complex non-linear relationships between users and items [27,28]. In the pursuit of implementing RSs, various DNN-based models have been proposed, with significant emphasis placed on devising sophisticated structures that can better accommodate user behavior data [29]. However, a notable difference between DNN-based models and the Latent Feature Analysis (LFA) model lies in their approaches to handling data [30,31,32,33,34]. While DNN-based models often operate on complete data, the observed entries within a user-item rating matrix, the reality is that RS-generated user-item rating matrices tend to exhibit low rating density [35,36,37,38]. This means that a significant portion of the matrix remains empty or contains missing ratings. Consequently, DNN-based models face challenges in effectively addressing the prevalent data sparsity issues in RSs [12,13,39,40].

Upon the aforementioned discussions, it becomes apparent that the LFA and DNN-based models have distinct advantages and disadvantages. Consequently, relying solely on either the LFA model or the DNN-based model cannot ensure optimal recommendation performance across diverse real-world application scenarios. To tackle this challenge, this study proposes a novel hybrid recommendation model called AutoLFA, which combines Autoencoder [41] and LFA techniques. The main concept behind AutoLFA is two-fold: (1) It leverages an Autoencoder and an LFA model separately to construct two distinct recommendation models, each residing in a unique metric representation space with its own set of strengths, and (2) it integrates the Autoencoder and LFA models using a customized self-adaptive weighting strategy, thereby capitalizing on the merits of both approaches. By incorporating elements from both the LFA model and DNN-based models, AutoLFA can deliver superior recommendation performance across various real-world application scenarios. This paper contributes to the field in the following ways:It proposes an AutoLFA model that aggregates the merits of both the LFA model and the DNN-based model by a customized self-adaptive weighting strategy;Theoretical analyses and model designs are provided for the proposed AutoLFA model;Extensive experiments on five real recommendation datasets are conducted to evaluate the proposed AutoLFA model. The results demonstrate that AutoLFA achieves significantly better recommendation performance than the related state-of-the-art models.

## 2. Related Work

Collaborative Filtering (CF) stands as a popular and effective approach for implementing an RS [2]. Its fundamental principle involves utilizing historical user behavior data to uncover similarities between users and items, thereby predicting users’ potential preferences for items. Matrix factorization serves as a prominent CF method, which typically maps the user-item rating matrix into two Latent matrices to explore the similarity between users and items [12]. Subsequently, the development of the LFA model introduced a notable distinction. Unlike matrix factorization, the LFA model exclusively trains the Latent Feature model using observed entries within the user-item rating matrix. As a result, LFA exhibits high efficiency and scalability, particularly in industrial applications [12,13]. Over time, several sophisticated LFA models have emerged, including those that consider data characteristics [42], incorporate non-negativity constraints [43], adopt generalized and fast-converging approaches [44], focus on smooth L_1_-norm regularization [12], employ probabilistic methods [45], apply dual loss [13], utilize prediction sampling [46,47], prioritize confidence-driven techniques [48], incorporate posterior neighborhood regularization [49], employ ensemble approaches involving multiple spaces and norms [50], explore graph regularization [51], and embrace deep structured architectures [52]. However, it is essential to note that the LFA model is inherently shallow and linear in nature. Consequently, it faces challenges when attempting to capture the deep non-linear relationships between users and items embedded within complex user-item rating matrices [21].

In recent times, Deep Neural Networks (DNN) have gained significant traction in the development of Collaborative Filtering (CF)-based RSs due to their powerful non-linear learning capabilities derived from deep learning structures [53]. DNN-based models aim to reduce the user-item rating matrix into a low-dimensional space to capture the similarities between users and items. A comprehensive review of DNN-based RSs was conducted by Zhang et al. [29]. Various sophisticated DNN-based models have emerged, including hybrid Autoencoder-based approaches [54], Autoencoder-based methods [41], multi-task learning-oriented techniques [11], graph neural network (GNN)-based models [55], neural factorization-based approaches [56], Autoencoders combined with radial basis function-based methods [57], attentional factorization-based models [58], hybrid deep models [28], biased Autoencoder-based techniques [21], and convolutional matrix factorization approaches [59]. However, it is worth noting that DNN-based models face challenges in addressing data sparsity problems since they are trained on complete data rather than solely relying on the observed entries within a user-item rating matrix [13]. Unfortunately, user-item rating matrices generated by RSs often exhibit very low rating densities.

Notably, although many LFA-based and DNN-based models have been built to achieve commendable recommendation performance, each approach has its own set of advantages and disadvantages. In comparison, the proposed AutoLFA is a hybrid recommendation model that combines the strengths of both Autoencoder and LFA techniques. This combination is controlled by a customized self-adaptive weighting strategy, ensuring that AutoLFA leverages the merits of both the LFA and DNN-based models, ultimately leading to superior recommendation performance across various real-world application scenarios.

## 3. Preliminaries

**Definition** **1**(**user behavior data**)**:** *Let M be a set of users, and N be a set of items. The matrix X ∈ ℝ with |M| rows and |N| columns records the interactions between different users and items. Here, x_mn_ represents the specific interaction specification of user m on item n. The vector* x*^m^ = {x_m1_, ⋯ x_m|N|_} denotes the behavioral data of user m across all items, while each item n can be represented as a vector* x*^n^ = {x_1n_, ⋯ x_|M|n_}. A binary matrix B ∈ ℝ with |M| rows and |N| columns distinguish the observed and unobserved interactions of X:*
(1)$$b_{mn}=\left\{\begin{array}{ll} 1 & \mathrm{if}\;x_{mn}\;\mathrm{observed}\\ 0 & \mathrm{otherwise} \end{array}\right.$$
*where b_mn_ denotes the specific entry on B.*

**Definition** **2**(**problem**)**:** *In recommender systems, two primary tasks exist: rating prediction and ranking prediction. Our proposed model is more suited to rating prediction, which aims to learn a parametric model denoted as f(·) using observed ratings of X in order to predict the unobserved ones. The prediction process can be represented as follows:*
(2)f(M,N;θ)→X.
*Here, θ represents the parameters of f(·). The objective function of f(·) is to minimize the empirical risk, expressed as:*

(3)
$$L(f)=\sum_{m\in M,n\in M}\epsilon \left(f(m,n;\theta ),x_{mn}\right).$$


*In this Equation, ϵ(·) denotes the error function that measures the distance between the predicted output $\hat{x}$_mn_ from f(m, n; θ) and the true rating x_mn_.*


## 4. The Proposed AutoLFA

As mentioned above, traditional approaches, such as Latent Feature Analysis (LFA), offer efficiency and scalability but may not capture complex non-linear relationships. On the other hand, deep neural networks (DNNs) show potential in capturing non-linear relationships but face challenges in dealing with data sparsity issues. Inspired by this finding, we propose AutoLFA with the aim of addressing both the challenge of LFA’s inability to capture complex non-linear relationships and the difficulty faced by DNN-based models in handling data sparsity issues. Figure 1 depicts the architecture of our proposed model, which can be separated into three steps: (1) Feed the user behavior data into the LFA-based and Autoencoder-based models separately; (2) obtain the predictions of the unobserved value from these two models; (3) aggregate the predictions of two models with a self-adaptive ensemble method to obtain final prediction $\hat{X}$. To illustrate the principle of Auto-LFA, we provide an example of predicting *x*_22_ in Figure 1. The predicted values from the two predictors differ by 3.5 in the LFA-based model and 2 in the Autoencoder-based model. These predictions are then weighted to derive the final prediction of 2.9. Next, we will provide a detailed description of AutoLFA.

### 4.1. The Latent Feature Analysis-Based (LFA-Based) Model

Given a user behavior matrix *X*, an LFA-based predictor aims to train two Latent Feature matrices *U* of size *|M| × d* and *V* of size *d × |N|* to generate the rank-d approximation $\hat{X}$ of *X* is based on the known entry of *X*, in which *d* is much smaller than *min*{*|M|*, *|N|*}. In this context, the row vectors of *U* represent user characteristics, while the column vectors of *V* represent item characteristics in the Latent Feature space.

We utilize the inner product space with an *L*_2_-norm $||\cdot |{|}_{L_{2}}^{2}$ as the *Loss* function in the LFA-based model to measure the distance between *X* and $\hat{X}$, as demonstrated below:(4)$$L(U,V)=\frac{1}{2}∥B⊙{\left(X-\hat{X}\right)}_{L_{2}}^{2}=\frac{1}{2}∥B⊙{\left(M-UV\right)}_{L_{2}}^{2}$$
where ⊙ denotes the Hadamard product. According to [12,13], regularization is crucial in preventing overfitting. By incorporating Tikhonov regularization into Equation (4), we obtain:(5)$$L(U,V)=\frac{1}{2}∥B⊙(X-UV){∥}_{L_{2}}^{2}+\frac{\lambda_{1}}{2}(\|U{\|}_{L_{2}}^{2}+\|V{\|}_{L_{2}}^{2}).$$ Here, *λ*_1_ is a hyperparameter that controls the intensity of its regularization penalty. It is worth noting that since the user cannot fully access all items leading *X* to be sparse, it is necessary to expand Equation (5) into a density-oriented form to improve efficiency, as follows [12,13]:(6)$$L(U,V)=\frac{1}{2}\sum_{x_{mn}\in X_{o}}{\left(x_{mn}-\sum_{k=1}^{d}u_{mk}v_{kn}\right)}^{2}+\frac{\lambda }{2}\sum_{x_{mn}\in X_{o}}\left(\sum_{k=1}^{d}u_{mk}^{2}+\sum_{k=1}^{d}v_{kn}^{2}\right).$$ Here, *u_mk_* represents the entry at the *u*-th row and *k*-th column of *U*, and *v_kn_* represents the entry at the *k*-th row, *n*-th column of *V*, and *X_o_* is the observed entries of *X*. We train the matrices *U* and *V* with the Adam optimizer [16] to obtain better prediction results.

### 4.2. The Autoencoder-Based Model

We chose the representative I-AutoRec [41] as the Autoencoder-based model. Formally, when given a user behavior data matrix *X*, I-AutoRec aims to solve the same problem as defined in Equation (3). The objective is to minimize the following loss function:(7)$$L(f)=\sum_{x^{n}\in M}\|\left(x^{n}-f\left(x^{n};\theta \right)\right)⊙b^{n}{\|}_{L_{2}}^{2}+\frac{\lambda_{2}}{2}\cdot (\|w_{1}{\|}_{L_{2}}^{2}+\cdots +\|w_{K}{\|}_{L_{2}}^{2}),$$
where *λ*_2_ > 0 represents the regularization factor to prevent I-AutoRec from overfitting. The parameter set *θ* = {*w*_1_, …, *w_k_*, *b*_1_, …, *b_k_*} includes the weighted terms *w_k_* and the intercept terms *b_k_* of the hidden layers, where *k* ∈ {1, 2, …, *K*}, **b***^n^* represents the n-th column of the index matrix *B*, and **x***^n^* corresponds to the item vector **x***^n^* = {*x*_1*n*_, …, *x_|M|n_*}.

### 4.3. Self-Adaptive Aggregation

Ensemble learning is a practical approach to combining multiple models. It is essential for the base models to exhibit diversity and accuracy [13]. To ensure diversity, we employ different types of models. Additionally, the representative LFA-based model and Autoencoder-based I-AutoRec ensure accuracy. As a result, the base models fulfil the two requirements for ensemble learning. To aggregate the models, we adopt a self-adaptive aggregation method based on their loss values on the validation set. The underlying principle is to increase the weight of the *t*-th base model if its loss decreases in the *i*-th training iteration or otherwise decreases. To comprehensively understand this idea, we will introduce relevant definitions to facilitate theoretical analysis.

**Definition** **3**(**Fractional *Loss* of Base Models**)**:** *The fractional loss of the t-th base model at the i-th iteration, denoted as Fl^t^(i), is computed as follows:*
(8)$$\begin{array}{c} Fl^{t}(i)=\sqrt{\sum_{m\in M,n\in N,(M,N\in \Gamma )}((x_{mn}-{\hat{x}}_{mn}^{t})\times m_{mn}{)}^{2}/{\left\|T\right\|}_{0}}\\ {\hat{x}}_{mn}^{t}=\left\{\begin{array}{ll} \sum_{k=1}^{d}u_{mk}\nu_{kn} & \mathrm{if}\;t=1\\ f(m,n;\theta ) & \mathrm{if}\;t=2 \end{array}\right. \end{array},$$
*where ||⋅||_0_ represents the L_0_-norm of a matrix which calculates the number of non-zero elements of it, and Γ is the validation subset of X.*

**Definition** **4**(**Cumulative *Loss* of Base Models**)**:** *We let Cl^t^(i) be the cumulative loss of Fl^t^ until the i-th training iteration and calculate as follows:*
(9)$$Cl^{t}(i)=\sum_{j=1}^{i}Sl^{t}(j).$$

**Definition** **5**(**Ensemble Weights**)**:** *The ensemble weight Ew^t^ for the t-th base model can be computed using the following formula:*
(10)$$Ew^{t}(i)=\frac{e^{-\delta Cl^{t}(i)}}{\sum_{l=1}^{2}e^{-\delta Cl^{t}(i)}}.$$
*Here, δ represents the equilibrium factor that controls the ensemble weights of the aggregation during the training process. Considering Definitions 3 to 5, the final prediction of AutoLFA in the i-th training iteration can be denoted as:*
(11)$${\hat{x}}_{mn}=\sum_{t=1}^{2}El^{t}(i)\cdot {\hat{x}}_{mn}^{t}.$$

### 4.4. Theoretical Analysis

The loss of the AutoLFA model at the *i*-th training iteration is represented as *Fl*(*i*) and computed as follows:(12)$$Fl(n)=\sqrt{\sum_{m\in M,n\in N,(M,N\in \Gamma )}((x_{mn}-{\hat{x}}_{mn})\times b_{mn}{)}^{2}/∥\Gamma {∥}_{0}},$$
where $\hat{x}$*_mn_* is calculated using align (11).

**Definition** **6**(**Cumulative *Loss* of AutoLFA**)**:** *The cumulative loss of the AutoLFA model is represented as Cl(i) and can be expressed as:*
(13)$$Cl(i)=\sum_{j=1}^{i}Fl(j).$$

**Theorem** **1.**
*For an AutoLFA model, assuming the Cl^t^(i) of the base models lies between [0, 1], and if Ew^t^(i) is set according to align (10) during training, the following alignment holds:*

(14)
$$Cl(I)\leq \min\{Cl^{t}(I)|t=1,2\}+\frac{\ln4}{\delta }+\frac{\delta I}{8},$$

*where I is the maximum iteration.*


By setting $\delta =\sqrt{1/\ln I}$ in Theorem 1, the upper bound becomes:(15)$$Cl(I)\leq \min\{Cl^{t}(I)|t=1,2\}+\ln2\sqrt{\ln I}+\frac{I}{8\sqrt{\ln I}},$$
where $\ln2\sqrt{\ln I}+\frac{I}{8\sqrt{\ln I}}$ is bound by *I* linearly. This leads us to the following proposition.

**Proposition** **1.**
*With $\delta =\sqrt{1/\ln I}$, the inequality holds:*

(16)
$$Cl(I)\leq \min\{Cl^{t}(I)|t=1,2\}+const,$$

*where the limit as I approaches infinity, const = 19.45.*


**Remark** **1.**
*Proposition 1 indicates that Cl(I) is constrained by min{Cl^t^(I) | t = 1, 2} + const, with $\delta =\sqrt{1/\ln I}$. Remarkably, each base variant with a different foundation allows them to exist in separate metric spaces. The ensemble weight in align (10) ensures that the AutoLFA model’s loss is always lower than the base models and benefits from the capabilities derived from the LFA and DNN-based models. Additionally, Proposition 1 is not intended to demonstrate the accuracy improvement of AutoLFA on the test set but rather to establish that the model possesses the advantages of the basic models. By showing that the proposed model achieves a smaller loss compared to each basic model used separately, it indicates that the model retains the respective strengths of the basic models without compromising its ability to fit the data.*


## 5. Experiments

In this section, we aim to address the following research questions (RQs) through subsequent experiments:RQ 1: Does the proposed AutoLFA model outperform state-of-the-art models in accurately predicting user behavior data?RQ 2: How does the AutoLFA model self-adaptively control the ensemble weights of its base models during the training process to ensure optimal performance?RQ 3: Are the base models of AutoLFA diversified in their ability to represent the same user behavior data matrix, thereby enhancing the performance of AutoLFA?RQ 4: What is the impact of the number of Latent Features and hidden units in the base models on the accuracy of AutoLFA?

### 5.1. General Settings

**Datasets**: For our experiments, we utilize five commonly used user-item datasets, as summarized in Table 1 These datasets include MovieLens_1M, MovieLens_100k, and MovieLens_HetRec from the MovieLens website, the Yahoo dataset from the Yahoo website, and the Douban dataset obtained from an open-access code. Table 1 summarizes the details of these datasets. The datasets are divided into train–validate–test sets using a ratio of 70%–10%–20%.

**Evaluation Metrics**: The primary objective of representing the user-item matrix is to predict missing ratings accurately. To assess the prediction accuracy of the tested models, we employ two evaluation metrics: root mean square error (RMSE) and mean absolute error (MAE), which are calculated according to [52].

**Baselines**: Our proposed MMA model is compared against seven state-of-the-art models: AutoRec (an original model), MF, and FML (Latent Feature Analysis-based models), and NRR, SparseFC, IGMC, and GLocal-K (deep-learning models). A brief description of these competing models is provided in Table 2.

**Implementation Details**: For all datasets, we set the learning rate to 0.001 for two models. We set the number of hidden units for the Autoencoder to 500 and the number of latent factors for the LFA model 30 to achieve better performance. The final testing results are obtained from the best-performing model, which exhibits the lowest prediction error on the validation set during training. The training process terminates when the preset threshold for training iterations is reached. All experiments are conducted on a GPU server with two 2.4 GHz Xeon Gold 6240 R CPUs, 376.40 GB RAM, and 4 Tesla V100 GPUs.

### 5.2. Performance Comparison (RQ. 1)

#### 5.2.1. Comparison of Prediction Accuracy

Table 3 presents the prediction accuracies of all models from D1 to D5. Statistical tests, including loss/tie/win analysis, the Wilcoxon signed-ranks test [60], and the Friedman test [21], are performed to analyze these results. The loss/tie/win analysis identifies cases where AutoLFA’s RMSE/MAE is higher/same/lower than other competitors. The Wilcoxon signed-ranks test is a non-parametric pairwise comparison method that determines if AutoLFA’s prediction accuracy is significantly higher than each comparison model based on *p*-values. The Friedman test compares the performance of multiple models across multiple datasets using F-rank values, with lower values indicating higher prediction accuracy. The comparative experiment results are normalized for better interpretation before conducting the Wilcoxon signed-ranks test and the Friedman test. The statistical analysis results of loss/tie/win, the Wilcoxon signed-ranks test, and the Friedman test are presented in the third-to-last, second-to-last, and last rows of Table 3. Key observations from Table 3 are as follows:AutoLFA achieves the lowest RMSE/MAE in most cases, with only ten cases of loss and one case of a tie in comparison. The total count of loss/tie/win cases is 7/1/62.All *p*-values are below the significance level of 0.1, indicating that AutoLFA outperforms all competitors in terms of prediction accuracy.AutoLFA obtains the lowest F-rank among all participants, confirming its highest accuracy across all datasets.

These observations highlight that AutoLFA achieves the highest prediction accuracy for predicting missing user data compared to other models.

#### 5.2.2. Comparison of Computational Efficiency

Figure 2 depicts the total time required for all participating models to reach the optimal RMSE on the validation dataset during training. The following observations can be made:LFA-based models generally exhibit higher computational efficiency compared to DNN-based models, as they are trained on observed user behavior data, unlike DNN-based models.Due to their complex data form and architecture, GNN-based models consume significant computational resources and time. From Figure 2, it is evident that IGMC surpasses 3000 s in time costs.Except for slightly longer time consumption on dataset D4, AutoLFA’s time consumption falls between LFA-based and GNN-based models. It is slightly higher than the original Autoencoder-based model but faster than other DNN-based models in most cases.

Based on these observations, we can conclude that the relatively simple structure of the base models in AutoLFA allows for acceptable total time costs after ensembling two base models.

### 5.3. The Self-Ensembling of MMA (RQ. 2)

To discuss the self-adaptive control of AutoLFA in ensembling different variant models and ensuring its performance, we monitor the variations of ensemble weights between its base models.

**Monitoring ensemble weight variations**: Figure 3 illustrates the changes in ensemble weights from D1 to D5, yielding the following observations:In most cases (e.g., Figure 3a–d), the ensemble weights of the Autoencoder-based model gradually increase and surpass the LFA-based model as the training progresses until the base models are fitted.In some instances, the LFA-based model’s weight may exceed that of the Autoencoder-based model. For example, in Figure 3e, the ensemble weight of the LFA-based model is greater than those of the Autoencoder-based model due to their faster convergence.

In conclusion, based on the experimental results and observations above, we can infer that AutoLFA effectively leverages different types of models. By aggregating these models in the ensemble stage, AutoLFA surpasses other state-of-the-art models in predicting missing ratings with only minor sacrifices in computational resources.

### 5.4. Distribution of Latent Features of Base Models (RQ. 3)

In order to investigate the diversity of the base models of AutoLFA and their abilities to predict the user behavior data matrix, we visually analyze the encoder output of the Autoencoder-based model, which represents the Latent Features of an Autoencoder model, and the Latent Features of the LFA-based model. The distribution of these Latent Features for the base models across all datasets is depicted in Figure 4. To analyze the distribution, we employ a Gaussian function and examine factors such as expectation (*µ*) and standard deviation (*σ*). The measurements of the full width at half maximum (FWHM) and the height of the Gaussian curve are also presented. From Figure 4, the following observations emerge:The distribution of Latent Features in the Autoencoder-based model tends to have more values concentrated at the extremes (i.e., 0 or 1), as shown in Figure 4f,h,i, while in the LFA-based model, the distribution tends to follow a normal distribution.After encoding, the Autoencoder-based model’s Latent Features are more likely to exhibit unusually high values within specific ranges. In contrast, in the LFA-based model, there are no extreme values, as depicted in Figure 4a,c,d.In some cases, the distribution of Latent Features in the Autoencoder-based model appears to be slightly more uniform compared to the LFA-based model, as illustrated in Figure 4e,j.

The observed information above indicates that the Autoencoder-based and LFA-based models have distinct representation characteristics, allowing AutoLFA to benefit from their different representation abilities. Consequently, AutoLFA ensures accurate prediction of missing ratings.

### 5.5. Influence of Numbers of Latent Features and Hidden Units to Base Models (RQ. 4)

We further investigate the impact of the number of Latent Features and hidden units in the base models on AutoLFA. Figure 5 illustrates the RMSE and MAE of AutoLFA as the number of Latent Features and hidden units varies simultaneously across D1 to D5. The following observations can be made from Figure 5:Increasing the number of Latent Features/hidden units from 2/20 to 20/300 results in a rapid improvement in the accuracy of AutoLFA. During this range, AutoLFA substantially increases accuracy without incurring significant computational costs.Once the number of Latent Features/hidden units reaches 25/400, the rate of accuracy improvement becomes less prominent in Figure 5b–d.

These observations suggest that setting the number of Latent Features/hidden units as 30/500 allows AutoLFA to achieve optimal accuracy in most cases without imposing significant computational resource demands. Although this setting may not yield the highest accuracy in certain cases, it remains relatively close to the optimal value.

## 6. Conclusions

This paper proposes a novel hybrid recommendation model by combining Autoencoder and LFA models, termed AutoLFA. Its main idea is two-fold: (1) It leverages an Autoencoder and a Latent Feature Analysis (LFA) model separately to construct two distinct recommendation models, each residing in a unique metric representation space with its own set of strengths, and (2) it integrates the Autoencoder and LFA models using a customized self-adaptive weighting strategy. As such, the merits of the LFA and DNN-based models are combined into the AutoLFA model, making it achieve superior recommendation performance under various real-world applications. The experiments investigate four research questions on five real recommendation datasets. The results verify that the proposed AutoLFA outperforms several state-of-the-art models. In the future, we plan to aggregate more variants of LFA-based and deep neural networks (DNNs)-based models to achieve better recommendation performance.

## Figures and Tables

**Figure 1 entropy-25-01062-f001:**
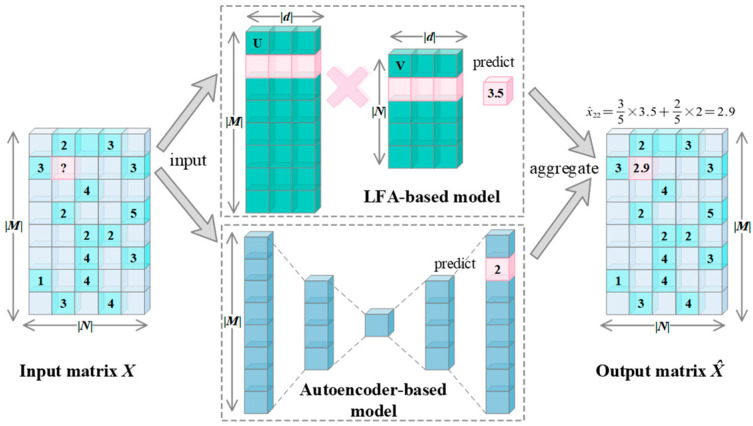
The architecture of the proposed AutoLFA model.

**Figure 2 entropy-25-01062-f002:**
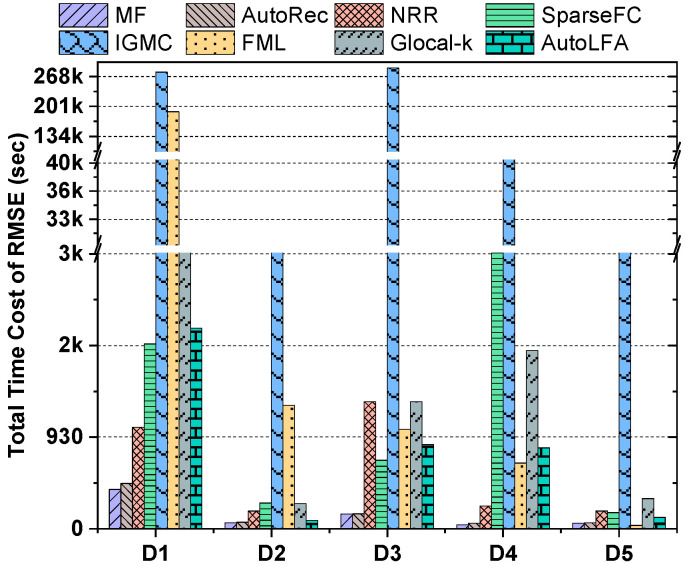
The histogram graph of the total time cost to reach the optimal accuracy of all the participating models.

**Figure 3 entropy-25-01062-f003:**
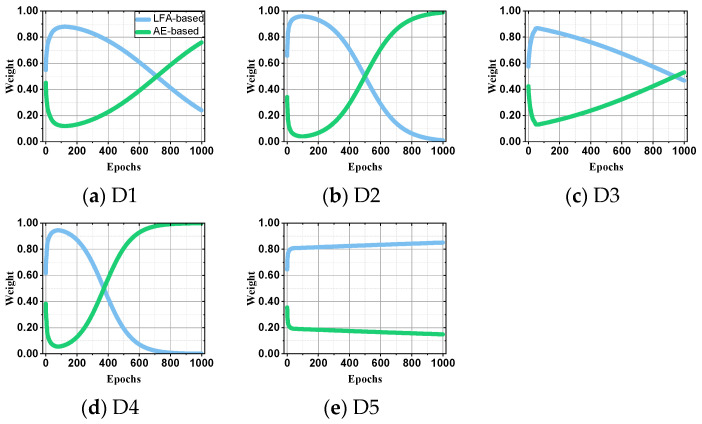
The changes in ensemble weights during the training process.

**Figure 4 entropy-25-01062-f004:**
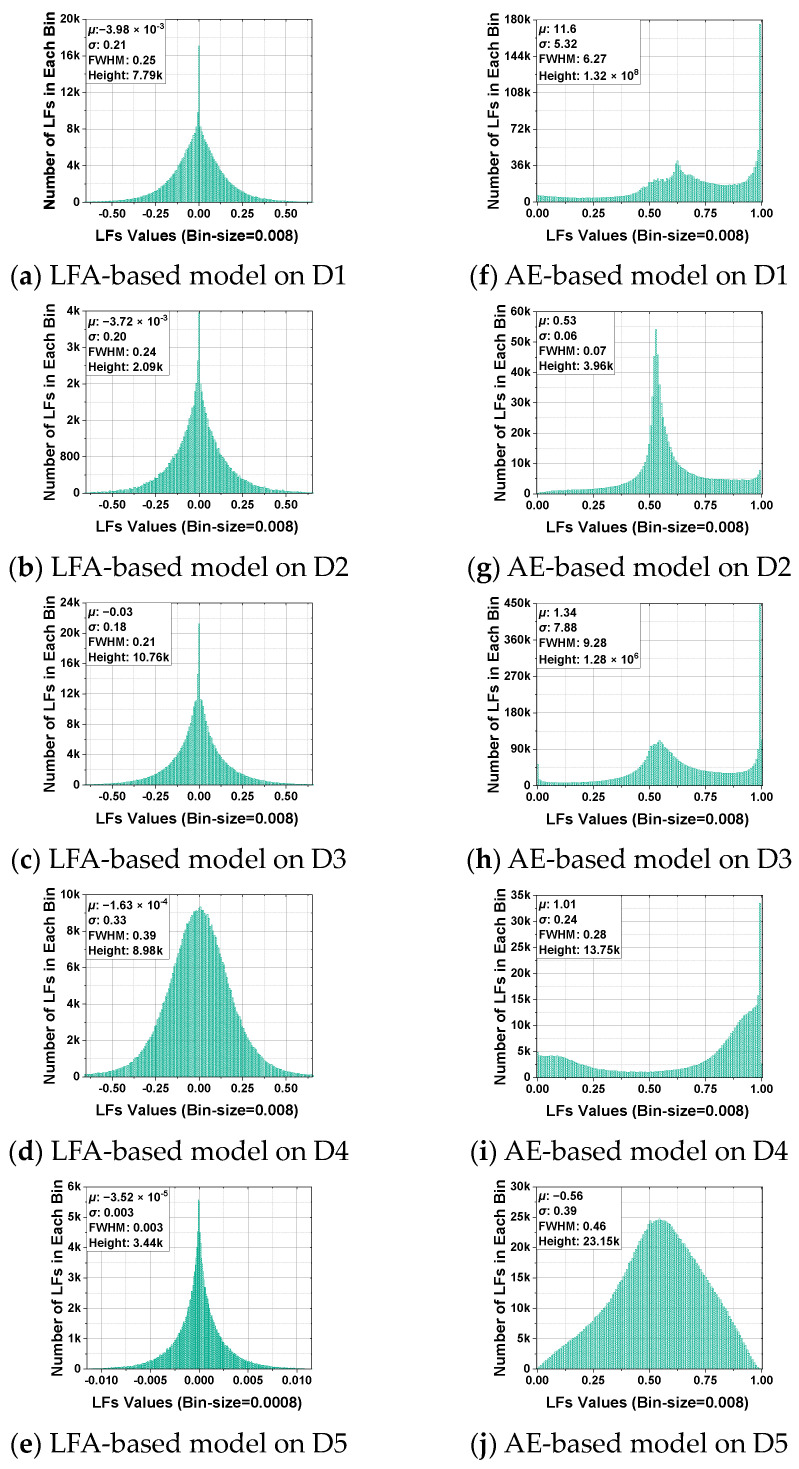
The distribution histogram of LFs of LFA-based and Autoencoder-based models from D1 to D5.

**Figure 5 entropy-25-01062-f005:**
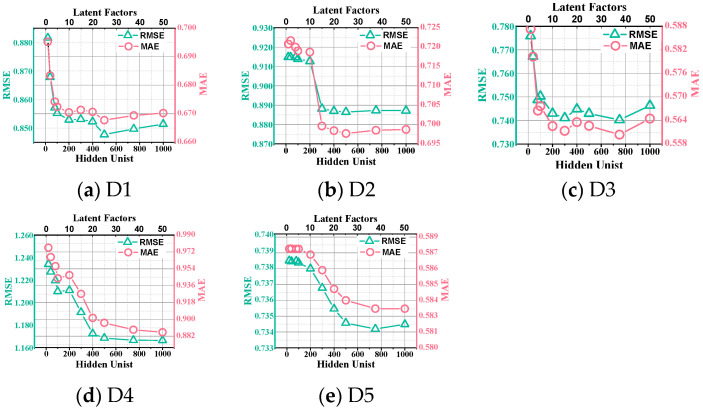
The line graphs of RMSE and MAE of AutoLFA from D1 to D5 as the number of Hidden Units and Latent Factors vary.

**Table 1 entropy-25-01062-t001:** Properties of all the datasets.

No.	Name	|M|	|N|	|HO|	Density *
D1	MovieLens_1M	6040	3952	1,000,209	4.19%
D2	MovieLens_100k	943	1682	100,000	6.30%
D3	MovieLens_HetRec	2113	10,109	855,598	4.01%
D4	Yahoo	15,400	1000	365,704	2.37%
D5	Douban	3000	3000	136,891	1.52%

* Density denotes the percentage of observed entries in the user-item matrix.

**Table 2 entropy-25-01062-t002:** Descriptions of all the contrasting models.

Model	Description
MF [10]	A representative LFA-based model for factorizing user-item matrix data in recommender systems. *Computer 2009.*
AutoRec [41]	A notable DNNs-based model for representing user-item data in recommender systems. *WWW 2015.*
NRR [11]	A DNNs-based multi-task learning framework for rating prediction in recommender systems. *SIGIR 2017.*
SparseFC [27]	A DNNs-based model that reparametrizes weight matrices into low-dimensional vectors to capture important features. *ICML 2018.*
IGMC [55]	A GNNs-based model for inductive matrix completion without using side information. *ICLR 2019.*
FML [9]	An LFA-based model that combines metric learning (distance space) and collaborative filtering. *IEEE TII 2020.*
GLocal-K [57]	A DNNs-based model for generalizing and representing user-item data in a low-dimensional space with important features. *CIKM 2021.*

**Table 3 entropy-25-01062-t003:** Performance comparison of AutoLFA and its competitors.

Dataset	Metric	MF	AutoRec	NRR	SparseFC	IGMC	FML	Glocal-K	AutoLFA
D1	RMSE	0.857 •	0.847 •	0.881 •	0.839 ∘	0.867 •	0.849 •	0.839 ∘	0.842
	MAE	0.673 •	0.667 •	0.691 •	0.656 ∘	0.681 •	0.667 •	0.655 ∘	0.664
D2	RMSE	0.913 •	0.897 •	0.923 •	0.899 •	0.915 •	0.904 •	0.892 •	0.887
	MAE	0.719 •	0.706 •	0.725 •	0.706 •	0.722 •	0.718 •	0.697	0.699
D3	RMSE	0.757 •	0.752 •	0.774 •	0.749 •	0.769 •	0.754 •	0.756 •	0.744
	MAE	0.572 •	0.569 •	0.583 •	0.567 •	0.582 •	0.573 •	0.573 •	0.562
D4	RMSE	1.206 •	1.172 •	1.227 •	1.203 •	1.133 ∘	1.176 •	1.204 •	1.167
	MAE	0.937 •	0.900 •	0.949 •	0.915 •	0.848 ∘	0.937 •	0.905 •	0.895
D5	RMSE	0.738 •	0.744 •	0.726 •	0.745 •	0.751 •	0.762 •	0.737	0.737
	MAE	0.588 •	0.588 •	0.573 •	0.587 •	0.594 •	0.598 •	0.580 ∘	0.584
Statistic	loss/tie/win	0/0/10	0/0/10	0/0/10	2/0/8	2/0/8	0/0/10	3/1/6	**7/1/62 ***
	*p*-value	0.0039	0.0039	0.0039	0.039	0.0195	0.039	0.0977	-
	F-rank	5.7	3.75	6.6	3.5	5.9	5.45	3.05	**2.05**

* The total loss/tie/win cases of AutoLFA. • The cases in which AutoLFA wins the other models in comparison. ∘ The cases in which AutoLFA loses the comparison.

## Data Availability

All data supporting the reported results in our study are publicly available and can be accessed through the following link: https://grouplens.org/datasets/movielens/, https://webscope.sandbox.yahoo.com/catalog.php?datatype=r, https://github.com/fmonti/mgcnn.

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
