# Peer review of "A Hybrid Recommender System Based on Autoencoder and Latent Feature Analysis"

_entropy, 2023, doi:10.3390/e25071062_

Round 1

Reviewer 1 Report

A recommender system is commonly used to extract valuable information from big data. Latent feature analysis (LFA) and deep neural networks (DNNs) are the two most adopted approaches to implement the Recommender system. However, both the LFA-based and the DNNs-based models cannot solely ensure optimal performance across diverse real-world application scenarios. To address this issue, this paper proposes the AutoLFA model which is a hybrid recommendation model by combining Autoencoder and Latent Feature Analysis (LFA) techniques. The pros and cons of this paper are as follows.

Pros:

1. Motivation. The motivation is clear. This paper aims at addressing the disadvantages of solely using LFA-based or DNNs-based methods. 

2. Methodology. The proposed AutoLFA model is novel and has technical depth. The theoretical analyses are detailed and well-formulated. 

3. Experimental Evaluation. Extensive experiments have been done on five datasets and the proposed AutoLFA model has been widely compared with seven state-of-the-art models. The results demonstrate AutoLFA's superior performance compared to state-of-the-art models. 

4. Comparative Analysis. Detailed discussions and analyses are provided for the experiments.

5. Presentation. This paper is easy to follow, conveying the proposed model and its contributions effectively.

Cons:

1. Language and Presentation: Improve language quality, address typos, and ensure accurate citation formatting. For example in Section 4 "Fig. 1 depict depict the architecture" there are two "depict".

2. It is better to provide more details regarding the experimental setup, including the used hyperparameters and optimization algorithms.

There are many grammatical and structural English language problems. Many sentences are written inappropriately.

Author Response

Response to Reviewer 1 Comments

Reviewer 1's Comments to the Authors: A recommender system is commonly used to extract valuable information from big data. Latent feature analysis (LFA) and deep neural networks (DNNs) are the two most adopted approaches to implement the Recommender system. However, both the LFA-based and the DNNs-based models cannot solely ensure optimal performance across diverse real-world application scenarios. To address this issue, this paper proposes the AutoLFA model which is a hybrid recommendation model by combining Autoencoder and Latent Feature Analysis (LFA) techniques. The pros and cons of this paper are as follows.

Author's Response to Reviewer 1: We sincerely appreciate Reviewer 1 for the valuable feedback on our paper. We are grateful for recognizing the pros of our work and for providing constructive criticism to help improve its quality. We have carefully considered the mentioned cons and addressed them in our revised manuscript. The modified content has been marked in blue in the revised version of the manuscript.

Point 1: Language and Presentation: Improve language quality, address typos, and ensure accurate citation formatting. For example in Section 4 "Fig. 1 depict depict the architecture" there are two "depict".

 Response 1: We sincerely appreciate Reviewer 1 for pointing out this problem. We acknowledge the need to improve the language quality, address typos, and ensure accurate citation formatting. We have thoroughly reviewed the paper and made the necessary corrections to enhance the language quality, eliminate typos, and ensure proper citation formatting throughout the manuscript. The modified parts has been marked in blue in the revised version of the manuscript.

Point 2: It is better to provide more details regarding the experimental setup, including the used hyperparameters and optimization algorithms.

Response 2: The authors sincerely thank the reviewer's insightful suggestion. In our revised version of the paper, we have comprehensively described the experimental setup, including the hyperparameters employed and the optimization algorithms utilized. This additional information will enhance the reproducibility and transparency of our research. The revised version of the manuscript highlights the modified content in blue.

Once again, we would like to express our gratitude to Reviewer 1 for the insightful comments. We have taken the suggestions seriously and made the necessary improvements to enhance the overall quality of the paper. We hope that the revised version will meet the expectations and contribute to advancing the field.

Reviewer 2 Report

In this paper, the authors first identify the limitations of existing Latent feature analysis (LFA)- and deep neural networks (DNNs)-related models and emphasize the significance of recommendation systems in handling large-scale data. In response to these challenges, they propose a hybrid recommendation system that combines Autoencoder and LFA techniques (called AutoLFA). Through rigorous experiments, the authors validate the effectiveness of the proposed AutoLFA model, as it outperforms seven state-of-the-art models on five real recommendation datasets. This paper provides a clear explanation of its motivation and demonstrates the feasibility of the proposed approach through experimentation. However, the article still has the following issues:

1. In the Introduction, it would be beneficial to provide additional explanation regarding the differences between the LFA and DNN models.

2. In the Related Work section, it would be helpful to include references to more recent works.

3. In the experimental section, it is recommended to provide the calculation methods for RMSE and MAE or include references to relevant sources.

4. There are some grammar errors in the article, please pay attention to improving them, for example, in section 4.2, the sentence "We choose the representative I-AutoRec [21] as the Autoencoder-based model." should be "We chose the representative I-AutoRec [21] as the Autoencoder-based model."

Author Response

Response to Reviewer 2 Comments

Reviewer 2's Comments to the Authors: In this paper, the authors first identify the limitations of existing Latent feature analysis (LFA)- and deep neural networks (DNNs)-related models and emphasize the significance of recommendation systems in handling large-scale data. In response to these challenges, they propose a hybrid recommendation system that combines Autoencoder and LFA techniques (called AutoLFA). Through rigorous experiments, the authors validate the effectiveness of the proposed AutoLFA model, as it outperforms seven state-of-the-art models on five real recommendation datasets. This paper provides a clear explanation of its motivation and demonstrates the feasibility of the proposed approach through experimentation.

Author's Response to Reviewer 2: We sincerely appreciate the valuable feedback provided by Reviewer 2 on our paper. These comments have helped us identify areas for further clarification and improvement. We have carefully considered and addressed each of the mentioned issues in our revised manuscript. The modified parts have been marked in blue in the revised version of the manuscript. The detailed responses are as follows.

Point 1: In the Introduction, it would be beneficial to provide additional explanation regarding the differences between the LFA and DNN models.

Response 1: The authors are thankful for Reviewer 2 pointing out this issue. In the revised version of the paper, we have expanded upon the distinctions between LFA and DNN models and approaches to recommendation systems in Section. The modified content is marked in blue in section 1 of the revised version of the manuscript. This clarification will provide readers with a better understanding of the motivations behind our proposed hybrid approach.

Point 2: In the Related Work section, it would be helpful to include references to more recent works.

Response 2: The authors are thankful for Reviewer 2 providing this insightful suggestion. We have thoroughly reviewed the latest literature and incorporated relevant references to ensure that our paper reflects the most up-to-date research in the field.

Point 3: In the experimental section, it is recommended to provide the calculation methods for RMSE and MAE or include references to relevant sources.

Response 3: The authors appreciate the reviewer's helpful suggestion. We have revised the experimental section to include a clear explanation of the calculation methods for RMSE and MAE by providing the appropriate references to ensure transparency and reproducibility. The modified parts in Section 5 have been marked in blue in the revised version of the manuscript.

Point 4: There are some grammar errors in the article, please pay attention to improving them, for example, in section 4.2, the sentence "We choose the representative I-AutoRec [21] as the Autoencoder-based model." should be "We chose the representative I-AutoRec [21] as the Autoencoder-based model."

Response 4: The authors greatly appreciate the reviewer for pointing out the grammatical errors in the paper. In the revised manuscript, we carefully examined the grammar errors mentioned and corrected the symbol mistakes as well. The modified content has been marked in blue in the revised version of the manuscript.

Once again, we sincerely thank Reviewer 2 for the insightful comments and suggestions, which will undoubtedly contribute to enhancing our paper. We have diligently incorporated these revisions and improvements to ensure the clarity, accuracy, and overall quality of the manuscript.

Round 2

Reviewer 2 Report

My comments and concerns have been addressed - I would like to support its acceptance. 

No more language comments